# The Eye as a Symbol of Ill-Fatedness in Two Canonical Picaresque Works: *Lazarillo de Tormes* and *Guzmán de Alfarache*

**Sarah Louise Ellis**

Department of Languages, Cultures and Film, University of Liverpool, Liverpool L69 7WY, UK; s.l.ellis@liverpool.ac.uk

**Abstract:** It seemed unimaginable that the eye, denoting visuality and deemed accurate and reliable in accordance with Aristotelian theories in circulation during the Spanish Golden Age could be considered as anything other than a revered hallmark of guidance and intellect. Nevertheless, the literary phenomenon) of the picaresque emerged at the onset of the seventeenth century to defy the chivalric and pastoral fantasies that were masking the real anxieties faced by an era of decline. The picaresque genre brought warning that turning a blind eye to Spain's already-waning fortunes could not last forever. Yet, by doing so, it lent favour to such blindness, underlining how the eye, both symbolically and substantially, actually evoked a sense of ill-fatedness and misfortune. This paper calls for an exploration of how an ominous utilisation of the eye is presented in the most canonical picaresque works: *Lazarillo de Tormes* and Mateo Alemán's *Guzmán de Alfarache*. From the imperative role of the blind man in opening the eyes of the young protagonist, to the doomed interpolated cosplay of seeing and unseeing throughout Lazarillo's trajectory, and from Guzmán's receptivity to appearances and Alemán's lending of visual lexicon to his picaro protagonist, one must ask: how and why does the bodily organ of the eye, through both notion and function, serve as a depiction of hardship and disaster within these picaresque texts, and how does it reflect the overarching societal views towards intellect and religion during this epoch of "ocularcentrism"?

**Keywords:** eye; vision; ocularcentrism; picaresque; *picaresca*; *pícaro*; *Lazarillo de Tormes*; *Guzmán de Alfarache*; Spanish Golden Age

## 1. Introductory Context

If early modern historian Stuart Clark maintains that the physical location of the eye emphasises its position as "the guide and ruler not just of the other senses but of the whole body" (Clark 2007, p. 10), then the same assertion of importance can be applied to the organ of the eye throughout the entirety of this article. From an intra-corporeal perspective, the optic connection to the brain, and, therefore, to the mind and realm of intellectualism, meant that sight was deemed the superior of the senses. Nevertheless, "while sight pertains to the function of the physical eye, vision involves the inner eye" (Stoichiţă 1995, p. 93), and, in an era so gripped by the prevalence of "ocularcentrism", both the inner and outer workings of the eye must be considered by way of an interdisciplinary approach in order to gain a complete understanding of just how esteemed this bodily organ really was during the Spanish Golden Age, and how and why such a fascination was captured by authors, particularly at this time.

Before delving into the myriad of ways in which the eye, both symbolically and substantively, was presented in Baroque Spanish literature—the picaresque in particular—we must examine more closely the conditions of such "ocularcentrism", summarised in the words of French theologian Jacques Ellul as "the privileging of vision, which can be traced as far back as the fourteenth-century Church's desperation in resorting to idolatry to maintain the trust of the faithful in a period of extreme crisis" (Jay 1988, p. 307). Once again, the notion of the eye as the most important sensory organ had become a pervasive

concept in Spain, not least in Europe[1], and its ubiquitous presence begged not only for the an updated examination of Ellul's ocularcentrist theory, whereby the links between the eye and idolatry had been ousted in exchange for connotations of vilification, but also for an extensive "re-thinking of the traditional separation between the visual and the verbal in studies of Iberian culture" (de Armas 2004, p. 7).

Generally speaking, the Baroque era in Spain was bookended by the rise and fall of the Spanish Golden Age; during which time, a myriad of literary and artistic forms, styles, and tropes were galvanised by the social instability and intellectual anxieties of the time. Philosophy and science had begun to establish their own procedures, moving away from the preconceptions of Christianity and the rigidity of traditional authorities (see Robbins 1998, p. 13), yet this was not to say that both religion and bureaucratic rule did not remain at the crux of such ideational values, or, at least, at the forefront of the author or artist's aspirations to conform to the jurisdiction of the Inquisition. Such a complex and uneasy amalgamation of innovating new ideas whilst complying to what was permitted by authorities thus led to cultural tensions, with even the literary and artistic realms sharing not only "affinities but rivalries too, [given] the problematic relationship between the ethical and the aesthetic" (Bergmann 2013, p. 116).

More than this, inter-literary and inter-artistic paradoxes were formed. From a literary stylistic perspective, this was delineated by hybridity: stories of romance laced with conflict, fables of morality mixed with parody, and tales of honour met with satire, ultimately resulting in the seriocomic form of the picaresque[2]. With regard to literary form, a unique moment whereby a fusion of orated tales and published epistles emerged, with such a "crossroad" demonstrating a swing towards written and printed works in favour of tales of verbal origin and distribution. This, in turn, served to elucidate the ever-increasing influence of a so-called Renaissance ocularcentrism in replacement of the importance of the voice and its auditory receptor, the ear, so often deemed the most steadily reliable sensory organ up until this point (see Calleja 1700, pp. 73–79). Concurrently, and, as this article will argue, significantly, it was during this aforementioned "crossroad of oral and written narrative" (Reed 1984, p. 34) that the genre known as the picaresque emerged, elaborately outlined as "the fictional confession of a liar which recounts in chronological order the orphaned hero's peregrinations from city to city and usually ends with either the defeat or the conversion of the inner man who both narrates and experiences the events" (Guillén 2015, p. 120). An interdisciplinary perspective of viewing the eye symbolically and substantially will thus be applied to the literary realm of the picaresque throughout this article, through the examination of the ominous utilisation and application of the eye in two of its most canonical works: *Lazarillo de Tormes* (1554) and *Guzmán de Alfarache* (1599–1604).

## 2. Deciphering the Eye in the Spanish Golden Age

With regards to early modern fascination surrounding vision, perception, and the organ of the eye itself, the reasons behind its notable ascent—particularly during the Spanish Golden Age—must be contemplated first and foremost, given that:

> The collapse of the "representational" model of vision based on species is exactly what happened in the 250 years between the early fifteenth and the late seventeenth centuries, when visual anomalies and paradoxes multiplied to such a degree that they overwhelmed the cognitive theory that permitted them to occur (Clark 2007, p. 20).

Thus, as an overarching understanding of the eye deepened, so too did awareness of how easily it might be deceived, given that "artistic advances with perspectival and anamorphic techniques testified to the eye's vulnerability to manipulation" (Woods 2019, p. 3). At the same time, such quarrelling forces of reformation and counter-reformation brought debates about idolatry to the fore, giving a renewed emphasis to long-standing fears that an admiring gaze might all too easily slip into blasphemy.

Stoichiţă points out that "until the last quarter of the sixteenth century, visions and the visionary experience interlinked with the function of the eye do not seem to have been the

particular preoccupation of Spanish artists or writers" (Stoichiţă 1995, p. 11), thus earmarking the onset of ocularcentrism at the latter stage of the Golden Age[3] and, henceforward, into the seventeenth-century, marking an era of uncertainty and decline in Spain. What combination of factors and influences led to the eyes' captivation and subsequent implementation in, above all, particular literary works pertaining to the picaresque genre during this timeframe? And why did authors of the picaresque make such negative connotations between both the function and the notion of the eye and the incidents recounted in their written texts?

The connection between eye and religion cannot be understated, even going back as far as the New Testament of the Bible, wherein the eye was beheld as "the lamp of the body" (Matthew 6:22), and the duality of things seen as "temporal" and things not seen as "eternal" are ruminated (Corinthians 4:18) to prefigure Ellul's argument that "the visual image produces an object outside the self solely there for our manipulation: claiming to represent the truth, vision actually operates on the level of deceptive artifice" (Jay 1988, pp. 309–10). Thus, the very dichotomy of the eye's function and perception is grounded in its religious underpinning, even if the *lux* versus *lumen* theories of optics went on to take a scientific dimension as more came to be understood about empiricism.

There is little doubt that the scientific and astronomical observations advancing in Europe at this time did much to arouse the curiosity of many, with one specific example being how the discovery of the New World, in which Spain played a central role, was reinforced by the Aristotelian doctrine on the sphericity of the Earth (see Ellis 2022, p. 7). Such profound searches for truth reinstated the eye's prominence in unearthing newfound discoveries and innovative breakthroughs—after all, Renaissance Europe was in unanimous agreement that the eye was fundamental in providing the most direct knowledge, demonstrating itself to be the organ of power, speed, liveliness, and accuracy (see Clark 2007, p. 10). This line of thought was also one trodden by theorists and philosophers, given that "the privileging of vision is central to theoretical approaches to the seventeenth century, which point to ocularcentrism as the motivation for [such] scientific and philosophical discourses that are seen as central to their modernity" (de Armas 2004, p. 151).

Nonetheless, it was not just the scientific and the systematic domains delineating the eminence of the eye, as the realm of the arts, namely that of writing, had a strongly visual component that was particularly palpable during the Spanish Golden Age, further cementing its imaginative prowess amidst a time of chiefly scientific and philosophical advancement. It would seem that poets and writers of fiction appealed to this sense, in particular, since it was thought that visualisation was a key to memory. Thus, the actions and images they were creating in their work would be remembered more easily by their readers (see de Armas 2004, p. 7).

Moreover, since Spanish literature of this period was highly visionary and more cerebral than it often appeared in hindsight, it could offer an extremely rich terrain for any research involving theoretical data on representation—namely "the problem of portraying the unrepresentable" (Stoichiţă 1995, p. 7), which is what this article sets out to establish by placing the literary phenomenon of the picaresque at its core. Given that the genre centres around a roguish protagonist—or pícaro—who tries unsuccessfully to carve a new life for himself, using trickery and ingenuity in frequent and outlandish attempts to cut the ties from his wretched past, only to bind himself further up in them, he is, then, symbolic of the unrepresentable side of society, the side we choose to turn a blind eye on. The more he tries to shake off his status as an outsider, the further he removes himself from any prospect of social acceptance, reinforcing "the fallibility of the eyes and the malleability of eyesight" (Clark 2007, p. 30) when it comes to the eye's efficiency and overall utilisation, whilst problematically presenting us with the eye as a perceived symbol of ill-fatedness, imploring its reconsideration as a tool no longer propelling us towards the gateway to divinity and intellect but instead to that of vice and immorality.

### 3. The Eye as a Symbol of Ill-Fatedness?

In order to fully gauge not only the significance of the eye but also, imperatively, its status as a symbol of ill-fatedness within the canonical doublet of picaresque works, *Lazarillo de Tormes* and *Guzmán de Alfarache*, a negative gaze must be cast upon its placement within these texts in line with the authors' views of cynicality at the conditions of the time. It is worth mentioning Cervantes' illustrious novel *Don Quijote* (1605) here, which was published following the success of Alemán's first instalment of *Guzmán*, and which embodies many similar tropes to its picaresque predecessors with regard to the deception of appearances and misapprehensions on worldly outlook. Nevertheless, the element of misguidance in *Don Quijote* can be attested more to the eponymous figure's alternate taxonomies of the world rather than by forging a direct link to ocularcentrism, whilst the text itself fails to exhibit the fundamentals of Guillén's picaresque classification, whereby he offers eight distinct features to ultimately delineate a definition of the genre[4]. Thus, it will find itself omitted from the main textual focus that this article prefers to align with: the "echt-picaresque"[5].

To contextualise its emergence, the picaresque itself was born as an alternative, as a bold response to idealistic ideas which instead could offer the radical (re)vision of the account of a man's life (see Sears 2003, p. 531), problematically blurring the lines between vision, form, and reform to leave readers with a fictional *testimonio* that appeared unsteady and inconsistent. This echoed the profoundly unsettling era in which it amassed its readers. If the picaresque had no blueprint and, therefore, no rigid literary framework to which writers could adhere, then neither did Spain's leaders when it came to the catastrophic decline of the nation. Thus, blindness sat at the helm, a theme that would prove crucial not only to the notion of being seen or unseen, but also to our understanding of a genre and a nation, "that threatens to unravel, to become something else entirely" (Sears 2003, p. 531). Blindness, it seemed, was figurative of the anxieties of being kept uninformed in a disturbing era of profound unsettlement (see Robbins 1998, p. 12).

In line with Matthew 6:23 of the New Testament, we can attest to blindness bearing overwhelmingly unfavourable overtones from a religious perspective. The eye being "bad" seemingly equates to a body "full of darkness", a judgement also poised by blind psychologist Georgina Kleege (1999, p. 71), who was keen to emphasise that "blindness inverts, perverts and thwarts all human relationships". Although this perception calls upon the physicality of being blind via visual impairment, more often than not in the picaresque, it is the act of choosing not to see—"to turn a blind eye"—that impinges upon the characters and their decisions and ultimately leads them to their downfall, be it religious, moral or both, thus proving that the eye's depiction as an omen of misfortune in the picaresque cannot pertain to blindness without also encountering active vision, even if this is then deliberately ignored.

The notion of sight through the eye gave rise to a wave of anxiety and scepticism during this epoch of ocularcentrism, as image became synonymous with external appearance rather than inward meaning, with representational truth masking its deceptive artifice through vision (see Jay 1988, p. 310). To some, this interlink between vision and vicissitude was further cemented by Biblical warnings of the eye as being a potential cause of self-undoing[6], as Clark (2007, pp. 24–25) summarises attitudes towards sight and vision at the time:

> The eye was the source of covetousness, adultery, idleness and pride, as well as things like excess and costliness of apparel; in each of these cases the sinful action is followed by a sinful perception. Thus, seeing comes between sin and the heart, and innocence is always compromised by sight.

Thus, taking into account the overarching ominous qualities of the eye that were surfacing at this time, an exploration of two of the picaresque's most revered works will now follow, calling into question the duality of the eye in serving both a symbolic and substantive function, which is symptomatic of the ill-fated fortunes of the pícaro protagonists it lends

itself to, and the wider conditions which they endured, utilising the confluence of blindness, sight, and insight considered as a singular, optical perspective (see de Armas 2004, p. 8) as an effective tool for such an ongoing interdisciplinary analysis.

## 4. The Eye in Lazarillo de Tormes

Guillén (2015, p. 72) offers a swift reminder that "no work embodies completely the picaresque". That being said, *Lazarillo de Tormes* undoubtedly determines a primitive starting point to the picaresque genre. It fell to this anonymously-published *novela* to skilfully integrate elements borrowed from a number of popular and literary sources into the pseudo-autobiographical perspective of this semi-credible account (see Bjornson 1977, p. 21). The written narrative, first published in 1554, recounts the life of a boy born into an impoverished family from the Salamanca slums as he "procura de ser bueno [tries to be good]" (Anon [1987] 2016, p. 22)—a template that he lives his life trying to adhere to, serving a number of masters and even working his way up to a post in office (albeit in the position of a lowly town crier) and residing with an archpriest. But he ultimately fails, instead facing a *caso* (case), which we see as the ostensible motive for his tale to *Vuestra Merced* (Your Honour), peppered with hypocrisy and laced with exaggeration.

How is the symbol of the eye, through its inner and outer workings, utilised in this text to encapsulate the ill-fatedness of the pícaro narrator, the society surrounding him, and even of the author of the text, who, still unbeknownst to the contemporary reader, cloaked his satirical dismay for contemporary Spain in the "seriocomic" form of this literary outlet?

From the very first line of *Lazarillo de Tormes*, the older protagonist reflecting on his life makes it clear that he is about to disclose details "nunca oídas ni vistas [never heard or seen before]" (Anon [1987] 2016, p. 3), with the importance of the reader's vision (and that of *Vuestra Merced)* immediately coming into play. It is worth mentioning here that the order of the noun form of the verb *oír* (to hear) preceding that of *ver* (to see) is not accidental if we are to consider how "hearing was more obviously an avenue of religious instruction and had been the sense of faith from the early Christian era onwards" (Clark 2007, p. 21). Lázaro is thus keen to demonstrate his apparently close ties to his faith upon initiating his narration, especially to the potential appeasement of *Vuestra Merced.* However, as his account unfolds, it becomes evident that it is Lázaro's obsession with seeing, unseeing, and the lack of seeing—or blindness—rather than hearing that comes to shape the entire *caso* that he puts forward.

Arguably, in hindsight, Lázaro is so preoccupied with how the account of his life is perceived by others that, by the end, he has actually blinded himself into believing his situation within society is that of "la cumbre de toda buena fortuna [the peak of all good fortune]" (Anon [1987] 2016, p. 135), when the reader can clearly see this, in fact, could not be further from the truth. We can discern that this self-blindness is at least partially influenced by the crucial portion of time young Lazarillo spent with the *ciego*, the blind man—his first master who guided him during his young but decisive adolescent years—an experience which unquestionably moulds the person he becomes thereafter as well as later on in life. Nowak (1990) goes one step further, so as to argue that the *ciego*, "endowed with an exaggerated ability to "see" in other ways" (Sears 2003, p. 538), prefigured the trajectory of Lázaro's life, therefore reiterating the ominous status of the function of the eye:

> As a Teiresian seer, he [the *ciego*] prophesises Lazarillo's condition in the seventh *tratado* (chapter) and anticipates Lazarillo's life with him through the seven episodes of the first *tratado* [...] The seven primary masters, in their respective chapters, will represent seven *exempla* of the seven capital sins. (p. 900).

A critical moment arises in Lazarillo's first journey as he ventures away from his hometown of Salamanca with the *ciego*, who teaches the pícaro protagonist his first lesson by reinforcing his unworldly naivety, violently slamming his head against a stone bull in jest of his childish behaviour. From this moment on, Lázaro recalls that "desperté de la simpleza en que como niño dormido estaba [I awoke from the foolish slumber I had sleepwalked through as a child]" (Anon [1987] 2016, p. 23). This episode significantly demonstrates that,

somewhat ironically, the young Lazarillo was taught how to truly see the world by a blind man, marking a seismic shift in the order of importance of the senses. Although Clark (see Clark 2007, p. 21) contemplates whether blindness enabled the individual affected (in this case the *ciego*) to be freed from peril and temptation, Kleege (1999, p. 51), in fact, is quick to stamp this view out, reminding that "blind men have the familiar foibles of sighted men, but the thing they lack, the thing that makes them different, also makes them potentially unruly, corruptible, dangerous."

　　Although Lazarillo eventually opts to leave his first master, the *ciego*, Sears (2003, p. 535) is quick to emphasize that "the problematic of seeing/not seeing" continues throughout the remainder of the *novela*, especially in the context of religious settings and when surrounded by religious figures, several of whom are served by Lazarillo: a *clérigo* (cleric), a *mercedario* (friar) and a *buldero* (distributor of papal bulls), as well as the *arcipreste* (archpriest) with whom Lázaro shares a house—and a wife. Unpacking the meaning behind the ill-omened function of the eye in these instances is especially significant if we contemplate how "early modern churchmen and other moralists warned of the inherent dangers in eyesight and of looking as a cause of wickedness" (Clark 2007, p. 24).

　　With the *clérigo*, Lazarillo has to resort to hiding in plain sight, given that "no pudiera cegarle [I could not blind him]" (Anon [1987] 2016, p. 57) to his trickery in attempting to open the bread-laden chest. The lack of details given regarding Lazarillo's time spent serving the *mercedario*, along with the implication of why he left so suddenly, "por esto y por otras cosillas que no digo [because of this and other reasons that I won't go into]" (Anon [1987] 2016, p. 111), turn this position on its head, as it is now Lazarillo who wishes to blind himself, or perhaps the older Lázaro is choosing not to (re)see certain events. Given that this latter episode alludes to something altogether more sinister[7], it is then up to the reader (not forgetting *Vuestra Merced*) to see for themselves, since it has become the narrator's decision to "hinder and exploit visionary experiences with their own interests in mind" (Stoichiţă 1995, p. 17).

　　In line with such exploitation of vision and sight, the *buldero*, Lazarillo's fifth master, teaches him deception as an art form, serving as an example that the vision of others can be altered by a third party for their own selfish benefit (usually to obtain some kind of financial profit). A dramatic and elaborate episode, which involves the *buldero* recruiting a fellow *aguacil* (bailiff) in order to stage a divine invention in which the papal bull can be seen as the extended hand of God, makes Lazarillo realise he is still susceptible to believing things too easily—"creí que ansí era, como otros muchos [I believed that it was (God) just as much as the others did" (Anon [1987] 2016, p. 123)—thus presenting the reader with a cyclical structure which links Lazarillo back to his previous naivety that he claimed he had shaken off as a child, and proving that the ill-fatedness of his child-like view of the world is still very much ingrained within his nature, even upon reaching adulthood, which, by this stage, poses to be more problematic given his ever-growing reliance on the function of the eye.

　　Despite the many lessons that Lazarillo is given in managing and manipulating vision and sight, whether it be his own or that of others, the doomed status of the eye and its manifold utilisations once again emerge at the end of the *novela* in the form of the *arcipreste*, who scolds Lázaro: "quien ha de mirar de malas lenguas, nunca medrará [he who looks out for gossip will never prosper]" (Anon [1987] 2016, p. 132). Instead of placing the verb *escuchar* (to listen) in this context, we are again presented with a verb interlinked with the use of eyesight in order to view the events which are unfolding, rather than use any other sensory function. By taking on this advice, Lázaro, through the use of his vision, has condemned himself to his ultimate situation of ill-fatedness. Then, as summarised by Sears (2003, p. 536):

> Seeing, according to the *arcipreste*, is therefore not believing. It is because Lázaro takes this advice to heart that he is able to conclude that with his second-hand clothes, his second-hand sword and his second-hand wife that *en este tiempo estaba en mi prosperidad* [at this time I found my prosperity].

Thus, Lázaro is firmly under the impression that he has excelled himself, and can strive no further for success, as he has already achieved it. His taking of the situation at face value, despite hearing constant rumours, is proof of his ultimate decision—be it conscious or not—to put sight, vision, and the overall function of the eye at the forefront of his sensory-ontological priority, whereby sight, as the sensual entity, comes prior to the rest of the senses, echoing the optometric work of Benito Daza de Valdés, whose pioneering ideas championing the excellence of sight coincided with the emergence of the picaresque (see Asorey-García et al. 2014).

## 5. The Eye in Guzmán de Alfarache

In short, *Guzmán de Alfarache* recounts the episodes from the life of the eponymous narrator, describing the misfortunes which befell him as he drifted between cities in Spain and Italy, "making a living in the employ[ment] of various masters or, when circumstances required, by begging, thieving, or gambling. Guzmán the narrator claims to have reformed himself and exchanged his wicked ways for a life of Christian righteousness" (Fritz 2018, p. 66), yet the authenticity of his repentance remains a subject of academic debate[8]. Thus construed as a by-product of Spain's decline at the turn of the seventeenth century, and favoured more than ever when the Kingdom of Castile was being ravaged by social plight and corruption (see Ellis 2022, p. 3), the publication of a picaresque work came at no more perfect moment than Mateo Alemán's *Guzmán de Alfarache*, which instantly found a mass readership[9], in part owing to the surge in access to printing and publication throughout Europe at this time, but especially given its surfacing in a new-found age of social disenchantment, the repercussions of which were to be reflected in the moral epistle at the helm of the picaresque literary phenomenon (see Sieber [1977] 2018, p. 7). How was the notion of the eye, when examined both symbolically and substantively, employed to demonstrate the ill-fated destiny of the archetypical pícaro and the backdrop of slow decline which he struggled to survive?

From the first *capítulo* (chapter) of Guzmán's long and morally-abiding account of his life, the theme of appearances and the deciphering of vision is instantly instilled into the reader: "aunque me tendrás por malo, no lo quisiera parecer [although you will get the wrong idea about me, it is not how it seems]" (Alemán [1987] 2012, p. 126). The use of words such as "enmascarar [to unmask]" and "robar a ojos vistas [to steal in plain sight]" accentuate the way Alemán boldly plays with the concepts of seeing and not seeing so early on in his work, with the choice of words undoubtedly foreshadowing the characteristics and actions that the pícaro protagonist will go on to inherit from his father. This is reiterated further by just how quickly Guzmán jumps to his father's defence: "no me podrás decir que amor paterno me ciega [I will not be told that I am blinded by my love for my father]" (Alemán [1987] 2012, p. 141). The use of the blindness concept here is paramount to the unfolding of Guzmán's account, as it is his obsession with trying to view an unyielding bond between himself and his *noble parentela* (noble patronage)—leaving his mother and their family home to make a name for himself and travelling to Italy to find his Genoese relations –which ultimately confine him to the doomed fate of that of a pícaro.

It is interesting to note that, unlike his pícaro counterpart Lazarillo, Guzmán does not undergo a period of servitude with a blind master, but the motif of blindness can still be encountered for different reasons. If viewed through the rhetorical lens of De Man (De Man [1971] 1983, p. xxix), the protagonist's recount "is neither heroic because it is not transforming, nor fatalistic because it does not submit or follow blindly, but it seeks to collaborate in an explicitation that is already at work and whose movement it attempts to espouse." One such way of interpreting this is through the infamous frontispiece image of a boatman resembling Guzmán, who is facing back to the shore of the picaresque narration, but rowing towards moral good. This imagery was used as a stark warning by Alemán, in fear that sightlessness on the reader's part would condemn them to misunderstanding his literary intentions and, instead, thinking that he, steered by his titular pícaro, wants to disembark on the land of the picaresque adventures (see Reed 1984, p. 67).

Dissimilar to his pícaro precursor Lazarillo, Guzmán is not forced to leave home to make something of himself, but does so due to "alentábame mucho el deseo de ver mundo [I was spurred on by my desire to see the world]" (Alemán [1987] 2012, p. 163), once again linking the function of eyesight to his ill-fated inevitability through his own actions. After setting off on his journey, his tears and the onset of darkness as night creeps in "no me dejaban ver cielo ni palmo de tierra por donde iba [prevented me from telling the sky apart from the ground—I could not see where I was going]" (Alemán [1987] 2012, p. 164). Ironically, this is arguably the part of Guzmán's journey where his mind is most clear, as, through not seeing, he arrives as the religious hospital of San Lázaro[10] to turn to help and exhibit his devout faith in God. Here, Alemán could be warning of the dangers involved in the visionary experience, which "allowed for the direct communication with the Sacred without the intermediary of the Church" (Stoichiţă 1995, p. 17), but it is evident that he is also reaffirming the eye's status as a mediator of ill-omens.

Another way in which the faltering of vision is dangerous, not only to Guzmán, but also his *curioso lector* (curious reader), is in the unreliability of his descriptive—bordering on hyperbolic—narration of appearances. Referring to his place of conception, San Juan de Alfarache, "está de frondosas arboleadas, lleno y esmaltado de varias flores. . . acompañado de plateadas corrientes, fuentes espejadas [is a place of leafy trees, adorned with various flowers . . . set amongst silver-plated streams and shimmering springs]" (Alemán [1987] 2012, p. 147). In other words, it is paradise on earth, thus likening Guzmán's parents and their act of forbidden intercourse to Adam and Eve committing the original sin. Such details call for Clark (see Clark 2007, p. 2) to voice his concern how literature at this time reverberated just how much the uncertainty and unreliability of the era tampered with the access to true visual reality, as optic disillusion fed off the disillusion it witnessed. After all, if "social reality is so completely founded on what can be seen, [then] the inhabitants of Guzmán's world pay little attention to anything else" (Folkenflik 1973, p. 347). Such scepticism of vision resonates with Friedman's "multiperspectivism"—a take on Spitzer's original theory of "perspectivism", whereby Alemán's literary inspiration, taken from the miscellany, is apparent, and it is emulated in the way his work can be critically approached. The interpolation of stories, descriptions, and, especially, the way "*Guzmán* conducts its narrative business in bold and protracted strokes" (Friedman 2015, p. 107) only furthers how discrepant the very vision the reader must rely upon appears.

Thus, when it comes to vision and eyesight in *Guzmán de Alfarache*, it becomes apparent that both are interconnected through the unreliability of the eye, the primary function of which is to deceive and to create a false sense of security intended to make the seer believe. By way of this picaresque account, not only is Guzmán the "seer", but so is the reader, which strikes a worrying chord by the time we close the book, given that:

> By the time we have learned the lessons of the courts and courtyards depicted by our narrator-hero, we readily concede that anyone who places his trust in the hollow appearances of this world is deceived by a mirage. It is, then, slightly disturbing for the reader to realize that when the narrator speaks of his own story—which is told to teach us the falsity of what we see—he very often does so in visual terms (Folkenflik 1973, p. 347).

## 6. Conclusions

Having examined the symbol of the eye in its contemporary context of seventeenth-century decline and literary brazenness through a case study of its utilisation within two of Spain's most revered literary texts, *Lazarillo de Tormes* and *Guzmán de Alfarache*, there is no doubt of the importance of the eye's multifaceted functioning—both symbolic and substantive—in emulating the ill-fated conditions pertaining to the picaresque genre and the characters ensnared within it.

The visual elements construed by the authors of these texts, whether through seeing, unseeing, or not seeing, lead us to understand the ocular process as one which is "unreal, imaginary and therefore personal, private and consequently, uncontrollable" (Stoichiţă

1995, p. 27). Thus, there is little wonder that the eye serves as the perfect symbol to define the ill-fatedness of those who live in a world constrained by appearances. If sight was the most noble and certain sense during this epoch of ocularcentrism, then the picaresque was crucial in also demonstrating it to be the most corruptible and most corrupting—an artistic backlash which reiterated the "necessary symmetry" of the dual conditions it emerged in and as a result of.

The picaresque serves as the perfect springboard for such illusory optic activity, on one level owing to its ability to paint a picture of the inevitable, yet slow and lamentable, Spanish decline through the encounters and misfortunes of its self-titular protagonists, but especially when recognising the fragmentary nature of the pseudo-autobiographical narrative, which is pieced together to attempt to construct a generic model, "along with the often-shadowy presence of the blind men, [who] are symptoms of a "visual" dissonance of which the authors of the picaresque are either unaware or refuse to see" (Sears 2003, p. 539).

We could go a step further in our analysis, aligning with Clark (see Clark 2007, p. 3), who ponders over the extent to which the power of the eye is influenced by "demonology", a disruption of the ocular-cognitive process which occurs when the devil enters the brain, or eye, or both to take control over the reception and perception of visual images—"tampering with the medium through which the visual species travelling, and altering the workings of the senses." However, given the religious undertones deemed imperative for such works to pass the Inquisitional tribunal at the time of their publication, it seems less likely that authors would endorse such a polemical theory throughout their narrative[11].

If the pícaro possessed the ability to blind himself to the things that happened in his illusory world, then the modern reader has not only adapted this skill but has learnt to manifest it to now yearn to see what the texts themselves do not. After all, perhaps we are actually the blind ones, "for not seeing exactly what Baroque readers found comforting about the picaresque" (Sears 2003, p. 543).

**Funding:** This research received no external funding.

**Institutional Review Board Statement:** Not applicable.

**Informed Consent Statement:** Not applicable.

**Data Availability Statement:** Data sharing is not applicable to this article. No new data were created or analyzed in this study.

**Conflicts of Interest:** The author declares no conflict of interest.

## Notes

1　Referring to Ellul's definition of ocularcentrism, if a period of "extreme crisis" is one of the root causes of the prevalence of this ideology, then there is little wonder that it once again gained traction at the turn of the seventeenth century, when uncertainty and degeneracy started to soar.

2　Blackburn (1979, p. 14) is the first to coin the term "seriocomic form" by way of introduction: "A picaresque novel is a seriocomic form that tends to appear at times when literary imagination is usually threatened by catastrophe: that is, when the very idea of existence commingles with the world of illusion."

3　In this article, I generally align with academics of theatre in that the Spanish Golden Age is considered to have fallen between 1550 and 1681, where 1681 coincides with the death of distinguished playwright Pedro Calderón de la Barca.

4　Claudio Guillén's (2015, pp. 77–106) eight characteristics of the picaresque can be sketched out in the following way: (1) Defining the pícaro (note that to him, this is most crucial), (2) A pseudoautobiographical frame or lens applied to the text, (3) An unreliable narrator, (4) A reflective pícaro—either morally, religiously, or both, (5) The stress on the pícaro's existence through material means, (6) The pícaro's awareness and subsequent mockery of different social classes and conditions, (7) The journey of the pícaro horizontally through space as well as vertically through society, and (8) the loosely episodic nature of the plot.

5　"Echt-picaresque" is a term coined by Barbara Fuchs (2021, p. 84), who identifies the works that come under this umbrella to be inherently picaresque (so, therefore, those which wholly align to Guillén's classification, as opposed to "para-picaresques" or complex, adjacent fictions).

6　In Matthew 5:29, giving in to the drawbacks of sight is alluded to as being a sinful choice and one that should be avoided at all costs: "If your right eye makes you stumble, tear it out and throw it from you; for it is better for you to lose one of the parts of your body, than for your whole body to be thrown into hell."

7  As footnoted by Ellis (2022, p. 26): "Lázaro's malicious code depends on readers' awareness of homosexual proclivities, real or alleged, among those of religious vocation"—such as buggery, sodomy, or even oral homosexual intercourse.

8  Although an intriguing line of enquiry, the authenticity of religion as it is upheld within Alemán's picaresque bestseller is not entirely relevant to this article. It can be explored further in the works of Carroll B. Johnson (1978) and Judith A. Whitenack (1985).

9  Incredibly, it is worth noting that Guzmán de Alfarache was one of the first genuine bestsellers in the history of printing, and by the time Alemán had penned la segunda parte in 1604, "twenty-six different editions and no less than fifty thousand copies had appeared in four or five years" (Guillén 2015, p. 143).

10  San Lázaro was a hospital founded in early modern Spain to care for leprosy patients. Sharing part of its name with Guzmán's pícaro prototype, Lázaro, begs the reader to consider how deliberate Alemán wanted this reference to be.

11  Stoichiţă (1995, p. 8) implores us to remember "the scale of surveillance on the part of the Inquisition in Spain (much more vigrous and strict than anywhere else [which] reflected a desire to control an imagery that was very often hidden from all institutionalized constraints".

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
