# Peer review of "The Eye as a Symbol of Ill-Fatedness in Two Canonical Picaresque Works: Lazarillo de Tormes and Guzmán de Alfarache"

_humanities, doi:10.3390/h12040077_

Round 1
Reviewer 1 Report
This is a well-written and structured article; to make it even better, specialized sources should be engaged, which in turn might help the discussion section. The article examines the role of vision during the Siglo de Oro, providing examples from two novels of the Spanish picaresque. About the focal theory, there is a reasonable objection: two of the authors mentioned (Clark, Ellul) suggest an ocularcentrism that does not exactly apply to the Spanish Golden Age, or maybe even the Baroque era. To write about vision during the Spanish Golden Age it seems the logical thing to do is deepen into how the eye and vision were considered at the time, also in relation to the rest of bodily senses, and those probably were more valued then than they are today. Ellul's opinions do not seem particularly germane to this discussion, but since they are already there, they might be contrasted, for instance, with those of a Spanish theologian from the time period at issue such as Diego Calleja: https://www.cervantesvirtual.com/obra/talentos-logrados-en-el-buen-uso-de-los-cinco-sentidos/. Calleja stresses hearing as "the best teacher of the soul", and he also dwells on the importance of the voice. Was Calleja's opinion part of the general consensus back then?
Good writers write for the ear, and during the Baroque Era, it was more even so, since they were read aloud. The works quoted in the essay suggest that (visual) appearances are not to be trusted (how does that apply to our current situation?); they stressed the deceptive quality of vision (do they suggest that maybe other senses are more reliable?). How does the ocularcentrism hypothesis apply to the Baroque Era, the Siglo de Oro, or the Spanish picaresque novel, especially considering that Guzman de Alfarache was a best-seller, translated into several languages?
English is fine, with minor typos (Jacques Ellul) or the use of contractions (couldn't).
Author Response
I'd like to start by thanking you for your thorough comments made on my manuscript piece. I especially agreed with the suggestions regarding broadening the critical context I applied to the focal theory within this piece, as well as addressing the eye vs. ear dilemma in a bit more depth. I went through the piece to fix the contractions and any spelling mistakes, and have tracked the changes on the corresponding document for your attention when following this piece up. Please note, given the time frame I was not able to address every single remark.

Reviewer 2 Report
The focus of the article is intriguing and productively chosen. There is an important attempt to connect the characteristic transformation of the themes of the eye, sight and blindness to the historical context. I think that the previous scholars' discussion on this transformation in the Spanish Golden Age context and more generally in the early modernity can be given a little more space, connecting the dots and explaining how Spain was special and how, in the author's opinion, this was reflected in the picaresque attitude to the eye. Perhaps even the paradox of a Golden Age, following the discovery of the New World and acquisition of new colonies, being at the same an age of decline should be addressed more explicitly. I think a more coherent and detailed, easier-to-follow narrative on the historical background will benefit the interdisciplinary readership of Humanities.
The final paragraph of section 2 seems to connect two very disparate points that could be distinguished and argued separately. The first one, on the Spanish literature of the period trying to portray the unrepresentable, seems particularly important and should probably be made more central and develop more consistently. I would ask, is the unrepresentable limited to the marginal, prohibited, sinful or ugly, as the paragraph explicitly says, or does it also include the abstract, subtle, complex and reflexive aspects of the human experience and social interaction? Speaking of the latter, shouldn't Don Quixote be mentioned at some point as picking up on some of the tendencies described, with regard to portraying the unrepresentable, as well as skepticism and suspicion about vision?
Also the sections dealing with each of the texts sound somewhat telegraphic, as if explicit connections between different quotes and points are withheld for lack of space. As it it, the article is far below the upper word limit, so there is ample opportunity to flesh out every important point.
Some endnotes could be profitably included in the main argument, to provide more context and connection. Quotes in general, whether from the primary or from secondary sources, need to be more explicitly commented on and tied into the argument in a clearer way.
I am attaching the pdf document with further comments and small technical suggestions in sticky notes.

An editing of English for run-on sentences, hard-to-follow sentences and occasional awkward expressions or word choice is recommended.
Author Response
I'd like to start by thanking you for your thorough comments made on my manuscript piece. I especially agreed with the suggestions regarding broadening the historical context on the Spanish Golden Age in general, as well as its call for an investigation into vision and eyesight. I went through the piece to elongate several points made and ensure that they flowed more eloquently than they did so previously. I used the PDF document that you so kindly provided me with to address and/or resolve other minor errors that cropped up in the article. I have tracked the changes on the corresponding document for your attention when following this piece up. Please note, given the time frame I was not able to address every single remark.

Reviewer 3 Report
The word contemporary (page 1, line 36) seems odd when applied to sixteenth-century Spain
Page 1 line 38 Jacques Ellul not jaques
just because a text announces itself as subversive, or is in a genre critics have previously seen of subversive, does not mean it is actually subversive. That point have to be argued and proven, as does what “subversive” means in the context.
The basic argument of the essay is that the theme of the ill-fatedness of the eye in Golden Age Spanish picaresque fiction is a negation or avoidance of an ocularcentrism associated, presumably, with Cartesian rationality, the Western epistemological tradition, and the male gaze. The argument is coherent and, insofar as one assumes its premises, well-elucidated. What I would aks the writer to do though is just to think about whether they are comparing two different registers: ocularcentrism as a tacit metaphysics view of knowledge, and the ill-fated eye as a stock, jocular reference in a far more colloquial literary genre. The essay assumes that we should be skeptical of ocularcentrism and champion the subversive, and that the picaresque is a subversive genre that challenges neat, rational constructions of how the world is. This all seme sto me contested. Ocularcentrism can underlie scientific views of the world, but also religious: certainly the mystical experiences of St john of the cross, as inward as they are, rely on ocularcentrism. Even in Don Quixote, the title character often misleads appearances, but this is not because ocularcentrism, as such, leads to his misapprehensions or can be blamed for his alternate taxonomies of the world. Also, the author suggests that reading the picaresque should make us value blindness more: but there is no reference to the Paul de Man of “Blindness and insight,” nor an awareness that such language is not necessarily sensitive to visually impaired people The assumptions behind the essay seem Bakhtinian 9though Bakhtin is not cited) and also too dependent on Martin Jay. Granted, Jay certainly wrote the authoritative book on visual hegemony in the western tradition, but it is a quarte-century old, the essential thinking behind it dates to the 1980s, and the dynamic, pro or con, with respect to ocularcentrism has moved forward; I a, surprised fro instance Juhani Pallasmaa is not cited. Seeing the picaresque as anti0ocularcentirc and subversive also tends to reaffirm stereotypes of Spanish culture as not intellectual and philosophical, going back to Spitzer’s ‘perspectivism’ and Auerbach writing about the Siglo de Oro in Mimesis. I would -like the author to just think about whether the opposition between the subversion of the picaresque and the hegemony of ocularcentrism is not a bit too binary.
That being said, the essay is able and well-educated and there is no reason
generally good
Author Response
I'd like to start by thanking you for your thorough comments made on my manuscript piece. I especially agreed with the suggestions put forward with regards to expanding the scope of critique used in my arguments, and found some interesting developments by way of Paul de Man and Spitzer's theory of Perspectivism especially. I have tracked the changes on the corresponding document for your attention when following this piece up. Please note, given the time frame I was not able to address every single remark.

Round 2
Reviewer 1 Report
This paper makes sense: it is well-written and well-referenced; it provides the reader with a fascinating discussion, a thoughtful conclusion, and the author has expanded and improved it considerably during the editing process. There might be a few typographical errors (lines 41, 42, 104, 266...) that a final spell check will quickly solve and make it perfectly publishable.
Only a few typographical errors have been detected.
Author Response
Thank you for checking my revised version so efficiently. All typographical and grammatical errors have been rectified, and references, notes etc. have all been checked thoroughly, so I am now satisfied to resubmit my piece.
